# RM-Line: A Ray-Model-Based Straight-Line Extraction Method for the Grid Map of Mobile Robot

Haoxin Liu * and Yonghui Zhang

School of Information and Communication Engineering, Hainan University, Haikou 570228, China
* Correspondence: haoxinliu@126.com

**Abstract:** This paper proposes a ray-model-based straight-line extraction method for the grid map of a mobile robot, call RM-Line. First, the edge map is obtained, with the help of the connectivity of the blank grid. Then, points containing complete line information, called active points, are obtained using a screening model. Lastly, a ray model is designed to extraction line segments. We evaluate the algorithm using the number of lines, the average distance from grids to the lines, and the running time. Experiments show that the proposed algorithm has better performance on grid maps compared to the state-of-the-art algorithms.

**Keywords:** line extraction; grid map; mobile robot; ray model





## 1. Introduction

The mobile robot maintains a grid map with the help of multi-sensor information. The grid map describes the distribution of obstacles and supports the mobile robot to achieve basic functions, such as obstacle avoidance, navigation, and positioning [1,2]. However, some more complex tasks, such as map compression [3] and map optimization [4,5], rely on deep information in grid maps, such as straight lines. Many studies in the field of mobile robots treated grid maps as images [6–8]; therefore, image-based methods are used to obtain straight lines from grid maps of mobile robots. As an important research direction in the field of robot vision, researchers have proposed numerous straight-line extraction methods, which are mainly divided into two categories: methods based on Hough transform and methods based on perceptual grouping [9].

### 1.1. The Hough Transform Approach

The Hough transform (HT) approach accumulate edges over the entire image into a histogram of potential line positions and orientations [10]. Accuracy can be improved by modeling uncertainty in local edges and propagating that uncertainty to the Hough map [11].

HT uses global information, allowing it to get a more comprehensive line, but it has some problems in identifying the endpoints of the line segment. Some methods look for the largest chain connecting or near connecting edges by analyzing the obtained straight lines [12,13]. There has also been some work to detect endpoints by analyzing features near peaks in the Hough map [14–18]. The main disadvantage of these methods is that, when there are multiple line segments on the line, only one segment will be detected.

On the other hand, the HT method is computationally intensive and prone to false positives in complex regions with dense edge responses [19]. To overcome the problem of false positives, the progressive probabilistic Hough transform (PPHT) [20] maps pixels to a parameter space with greater probability, which reduces false positives to a certain extent. Yang et al. introduced orthogonal image scanning to reduce the computational complexity of HT [21]. Almazan proposed the Markov chain marginal line segment detector. In the first stage, straight lines are detected using the global probabilistic Hough method. In the

second stage, the points on the line segments are modeled as Markov chains, and then the optimal probabilities are calculated using standard dynamic programming algorithms. This method improves the accuracy and completeness of line segments [22]. However, these methods have not yet completely solved the problem of HT's large computational load and easy false positives in complex regions.

### 1.2. Perceptual Grouping Approach

The perceptual grouping approach groups roughly collinear local features into initial line segments by analyzing pixel information, such as gradient, proximity, and connectivity, and then evaluates the initial line segments using appropriate criteria.

Boldt et al. proposed a hierarchical heuristic framework [23], on this basis, Nieto et al. proposed the slice sampling weighted mean shift (SSWMS) algorithm [24], which iteratively selects and grows pixels according to the gradient structure. The biologically inspired method [25] first uses "simple units" to detect local structures, then integrates these responses with "complex units", and finally detects endpoints with "hyper-complex" mechanisms.

Gioi et al. proposed a fast line segment detector (LSD) with a false detection control [26], which sorts the pixels according to the gradient, then extracts line segments according to the gradient direction of the pixels in the local area. They used the Helmholtz principle to judge the extraction results, which transfers the threshold to a quantity that is much easier to set rationally. They improved the algorithm in a subsequent article [27], aiming to process any digital image without parameter tuning. Compared with HT, LSD has certain advantages in operation speed and precision. However, LSD cannot guarantee the accuracy of the results when the local region consists of pixels with similar gradient magnitudes. To this end, Akinlar et al. proposed edge drawing line segments detection (EDLines) [28], which first performs edge detection and constructs pixel chains, and then obtains initial line segment through the least-squares method in a local range. EDLine improves the extraction effect in local regions with similar gradient magnitudes; however, in edge detection, the threshold of the operator is difficult to select and needs to be corrected according to the specific image [29]. Lu et al. proposed CannyLines [30]. They designed a parameter-free Canny operator named CannyPF to robustly extract edge maps from input images by adaptively setting the low and high thresholds of the Canny operator. Then, clusters of collinear points are collected from edges using edge linking and splitting techniques. Finally, the method based on least squares is used to fit the initial line segment. CannyLines improves edge detection. Although simpler, these methods are not robust to gaps or intersections in edge maps [31].

### 1.3. Our Approach

Our work targets grid maps for indoor mobile robots. The grid in the map has three possible values: OBSTACLE = 0, which means that there are obstacles in the grid; BLANK = 1, which means that there are no obstacles in this grid; UNKNOWN = 2, which means that the grid is not detected.

When the existing HT methods are used to extract straight lines from grid maps, the computational load is large. When the existing perceptual grouping methods are used to extract straight lines from grid maps, there is a certain distance between the obtained straight line and the real obstacle contour due to less local information (the value of the pixel has only three discrete values).

This paper proposes an algorithm for extracting straight lines from grid maps, called RM-Line. The following terms are used in the algorithm:

1.　Connected domain. The connected blank grids in the grid map are clustered, and the area formed by the grids in the cluster is called the connected domain;
2.　Edge point. The obstacle grids located at the edge of the connected domains;
3.　Active points. The obstacle grids that belong to a line.

RM-Line first uses the connected domain of the grid map to obtain edge points, and then uses the screening model to obtain active points. Lastly, a ray model is used to derive line segments.

### 1.4. Organization

The remainder of the article is organized as follows:

Section 2 presents the materials and methods. Section 2.1 describes the acquisition of the edge map. Section 2.2 describes the acquisition of active points, i.e., points containing complete line information. Section 2.3 introduces the ray model and the acquisition of line segments. Section 2.4 introduces the flow of the algorithm.

Section 3 presents the experiments and results. Section 3.1 introduces the evaluation criteria. Section 3.2 presents Experiment 1, which is in a single room. Section 3.3 introduces Experiment 2, where the environment consists of two rooms. Section 3.4 describes Experiment 3, which was performed in a corridor.

Section 4 provides the conclusions and future work.

## 2. Materials and Methods

### 2.1. Edge Point Extraction Method Based on Connected Domain

The line information in the grid map of the mobile robot is hidden in the contour of the obstacles, and the pixels inside the contour increase the search range of the line extraction algorithm and may cause interference. Therefore, this paper proposes a method based on connected domains to extract edge points from the grid map, thereby narrowing the search range of the line extraction algorithm.

Figure 1 shows the extraction process of the edge points.

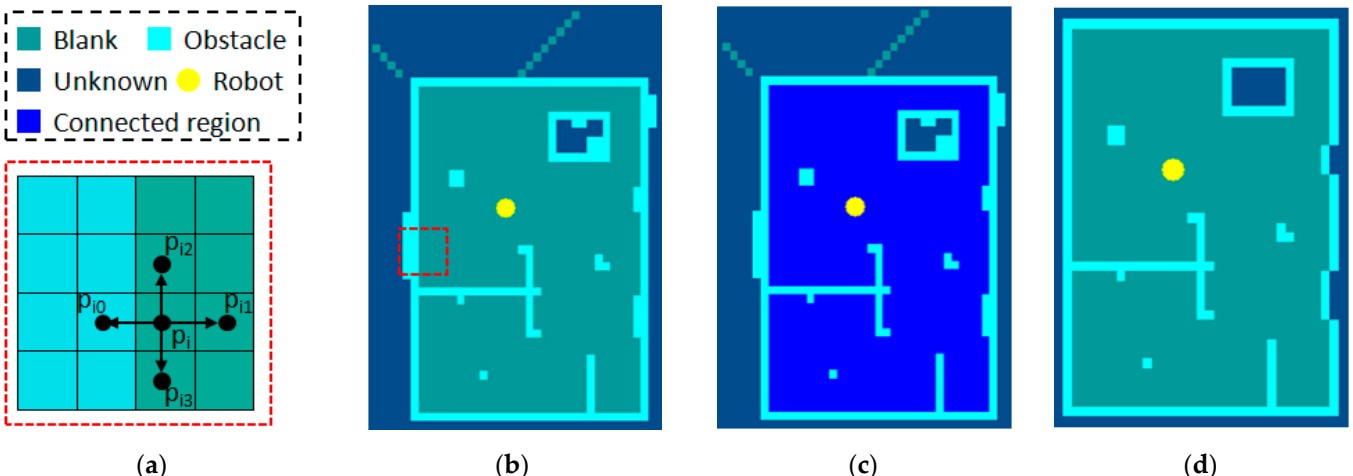

**Figure 1.** The extraction process of the edge points: (**a**) four neighbor grids of the BLANK grid $p_i$; (**b**) grid map of the mobile robot; (**c**) the connected domain; (**d**) the edge map.

(1) Figure 1b shows a grid map of the mobile robot used as input, where the yellow dot is the robot, the turquoise points are the BLANK grids, the cyan points are the OBSTACLE grids, and the steel-blue points are the UNKNOWN grids.

(2) Create an auxiliary map, denoted as *AuxMap*, whose initial value is 0. Create a cache array, denoted as *OpenArray*; let $Array[0] = p_{rob}$, where $p_{rob}$ is the position of the robot.

(3) Figure 1a shows the grid $p_i(x_i, y_i)$ in the OpenArray; search for its neighbor point $p_{ik}$, where $p_{ik} \in \{p_{i0}(x_i + 1, y_i), p_{i1}(x_i - 1, y_i), p_{i2}(x_i, y_i + 1), p_{i3}(x_i, y_i - 1)\}$. If $p_{ik}$ is a BLANK grid, and $AuxMap(p_{ik}) == 0$, then add $p_{ik}$ to OpenArray and let $AuxMap(p_{ik}) = 1$. If $p_{ik}$ is an OBSTACLE grid, let $AuxMap(p_{ik}) = 2$, but do not add it to *OpenArray*.

(4) Repeat (3) until all points in the *OpenArray* are processed.

(5) Grids with *AuxMap(p)* = 1 form a connected domain, as shown in the blue part in Figure 1c. Grids with *AuxMap(p)* = 2 are the edge points, as shown in the cyan part in Figure 1d.

It can be seen from Figure 1 that the edge point extraction process filters out the points inside the contour or outside the map, to reduce the search range of the line extraction algorithm.

### 2.2. Active Point Extraction Method Based on Screening Model

Edge points describe the contour of the obstacle; however, there are some edge points, such as isolated points, that do not contain line information. Moreover. there are other edge points, such as the point at the end of the contour, which only contain part of the line information. Taking these points as the starting point of the extraction line may yield unsatisfactory results; hence, this paper proposes a screening model based on the distribution of local pixels and uses the screening model to extract active points from the edge map, to provide a better starting point for the line extraction algorithm.

The screening model is shown in Figure 2. Figure 3 shows the active point extraction process based on the screening model.

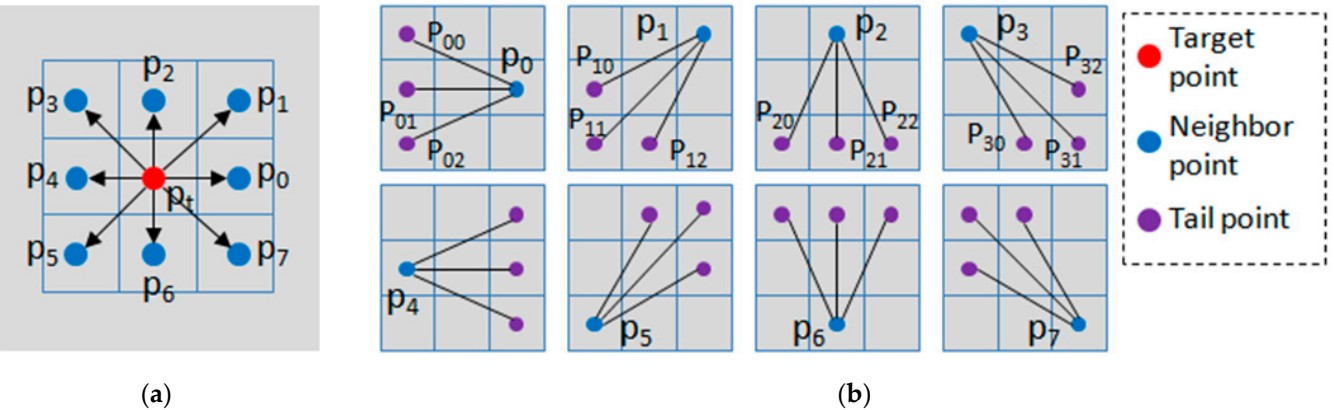

**Figure 2.** Screening model. (**a**) The target grid $p_t$ has eight neighbor points, denoted as $\{p_0, p_1, \ldots, p_7\}$. (**b**) Each neighbor point has three tail points, denoted as $\{p_{i0}, p_{i1}, p_{i2}\}$.

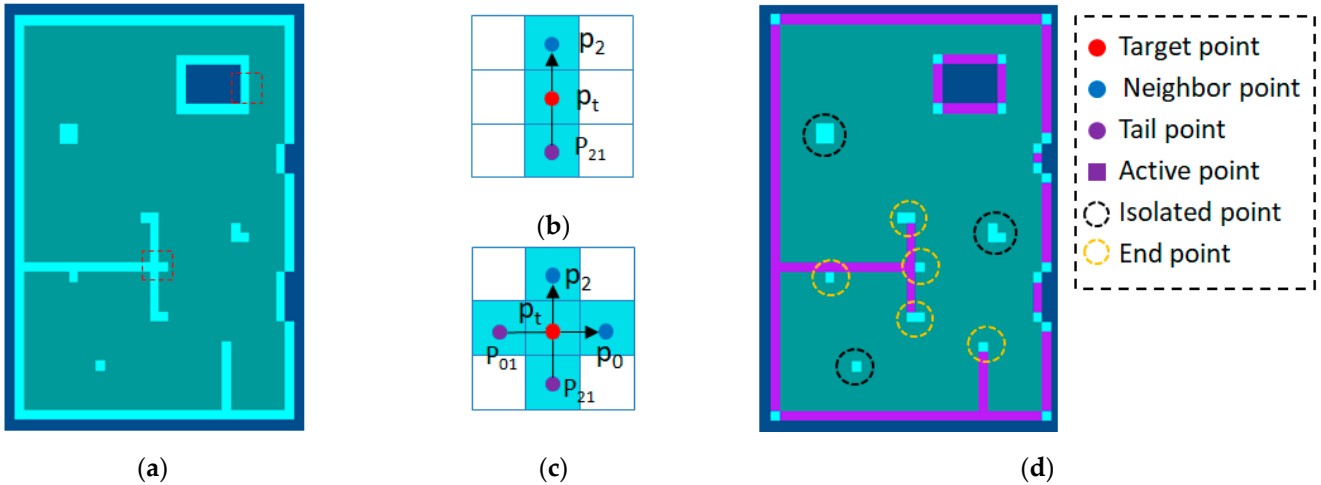

**Figure 3.** (**a**) Edge map; (**b**) a grid in the top of the edge map; (**c**) a grid in the center of the edge map; (**d**) active grids.

(1) Figure 3a shows the edge map used as input.

(2) For the edge point $p_t$, if one of its neighbor points $p_i$ is an OBSTACLE grid, and at least one of the three tail points of $p_i$ is an OBSTACLE grid, then $p_t$ will be marked as an

active point. Taking Figure 3b as an example, the neighbor point $p_2$ of the target point $p_t$ is an OBSTACLE grid, and the tail point $p_{21}$ is also an OBSTACLE grid; thus, $p_t$ is determined as an active point.

(3) Repeat (2) until all edge points are processed.

(4) The final result is shown in Figure 3d.

It can be seen from Figure 3 that points at the end of the contour (indicated by black circles in Figure 3d) and isolated points (indicated by yellow circles in Figure 3d) are regarded as inactive points, which do not contain complete line information. On the contrary, the extracted active points contain complete line information, which is beneficial to the line extraction algorithm to obtain better results.

### 2.3. The Ray Model

In the above sections, edge points were extracted, and active points were marked. Next, we propose a ray model, combined with active points, to extract straight lines from the edge map.

The ray model is shown in Figure 4.

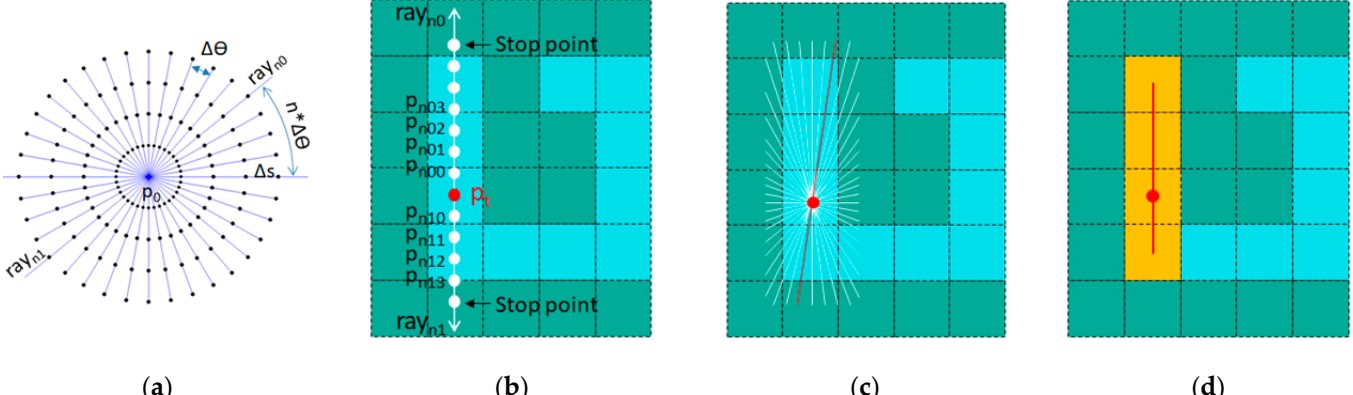

**Figure 4.** (**a**) The ray model; (**b**) scoring points; edge map; (**c**) the best ray; (**d**) line segment.

As shown in Figure 4a, $p_0(x_0, y_0)$ is taken as the starting point and $n \times \Delta\theta$ is taken as the direction to get the ray $ray_{n0}$ and the reverse ray $ray_{n1}$, where $\Delta\theta$ is the set angle interval, and $0 \leq n \leq \frac{180}{\Delta\theta}$. As shown in Figure 4b, a series of scoring points are set at intervals of $\Delta s$ on $ray_{n0}$, denoted as $\{p_{n00}, \ldots, p_{n0k}\}$. Similarly, scoring points are set on $ray_{n1}$, denoted as $\{p_{n11}, \ldots, p_{n1m}\}$, where $\Delta s$ is the set distance interval, $k \geq 0$, and $m \geq 0$.

As shown in Figure 4b, the target grid $p_t$ is used as the center of the ray model. For $ray_{n0}$, its score is calculated starting from the scoring point $p_{n00}$. If the grid corresponding to the scoring point $p_{n0k}$ is an OBSTACLE grid, the score of $ray_{n0}$ is increased by 1, and the calculation continues. If it is not an OBSTACLE grid, the calculation is stopped, and the current scoring point $p_{n0k}$ is noded as the stop-point of $ray_{n0}$. Similarly, the score of $ray_{n1}$ is calculated starting from the scoring point $p_{n10}$. The score function of the ray is shown in Equation (1).

$$score_n = \sum_{k=0}^{e0} S(p_{n0k}) + \sum_{m=0}^{e1} S(p_{n1m}), \tag{1}$$

where $0 \leq n \leq \frac{180}{\Delta\theta}$; $e0$ is the label of the stop-point of $ray_{n0}$, and $e1$ is the label of the stop-point of $ray_{n1}$; $S(p_{n0k})$ is the score of $ray_{n0}$ and $S(p_{n1m})$ is the score of $ray_{n1}$. $S(p)$ can be obtained using Equation (2).

$$s(p) = \begin{cases} 1 & map(p) == OBSTACLE \\ 0 & others \end{cases}. \tag{2}$$

For the target grid $p_t$, the scores of all rays are calculated, and $ray_{k0}$ and $ray_{k1}$ with the highest scores are selected as the output results, as shown by the red line in Figure 4c.

These grids that are traversed by $ray_{k0}$ and $ray_{k1}$ form a cluster, as shown by the orange grids in Figure 4d.

The clustered grids are fitted to a straight line using a least squares-based method, as shown by the red line in Figure 4d. $\{p_0(x_0, y_0), \ p_0(x_0, y_0), \ldots, \ p_N(x_N, y_N)\}$ are fitted into a straight line using Equations (3) and (4), with the straight-line form $ax + by + c = 0$.

$$\begin{cases} \mu_x = \frac{1}{N}\sum_{i=1}^{N} x_i \quad \mu_y = \frac{1}{N}\sum_{i=1}^{N} y_i \\ \mu_{xx} = \frac{1}{N}\sum_{i=1}^{N}(x_i - \mu_x)^2 \\ \mu_{yy} = \frac{1}{N}\sum_{i=1}^{N}(y_i - \mu_y)^2 \\ \mu_{xy} = \frac{1}{N}\sum_{i=1}^{N}(x_i - \mu_x)(y_i - \mu_y) \end{cases} , \tag{3}$$

$$\begin{cases} a = \frac{\mu_{xy}}{\sqrt{\mu_{xx}^2 + \mu_{yy}^2}} \\ b = \frac{-\mu_{xx}}{\sqrt{\mu_{xx}^2 + \mu_{xy}^2}} \\ c = -a\mu_x - b\mu_y \end{cases} , \tag{4}$$

where $\mu_x$ is the average of the *x*-coordinates of all points, $\mu_y$ is the average of the *y*-coordinates, and $N$ is the number of grids. $\mu_{xx}$ is the mean squared error of the *x*-axis, $\mu_{yy}$ is the mean squared error of the y-axis, and $\mu_{xy}$ is the covariance. (*a,b,c*) are the parameters of the line, which has the form $ax + by + c = 0$.

### 2.4. The Algorithm Flow of RM-Line

Figure 5 shows the process of RM-Line. We use a simulated map here to demonstrate the computational process of the algorithm.

(1) As shown in Figure 5a, the grid map is used as input.

(2) As shown in Figure 5b, edge points are obtained using the method based on the connected domain.

(3) As shown in Figure 5c, active points are obtained using an approach based on a screening model.

(4) Start searching for all edge points. Let $x = 0$, $y = 0$, and mark all active points as free.

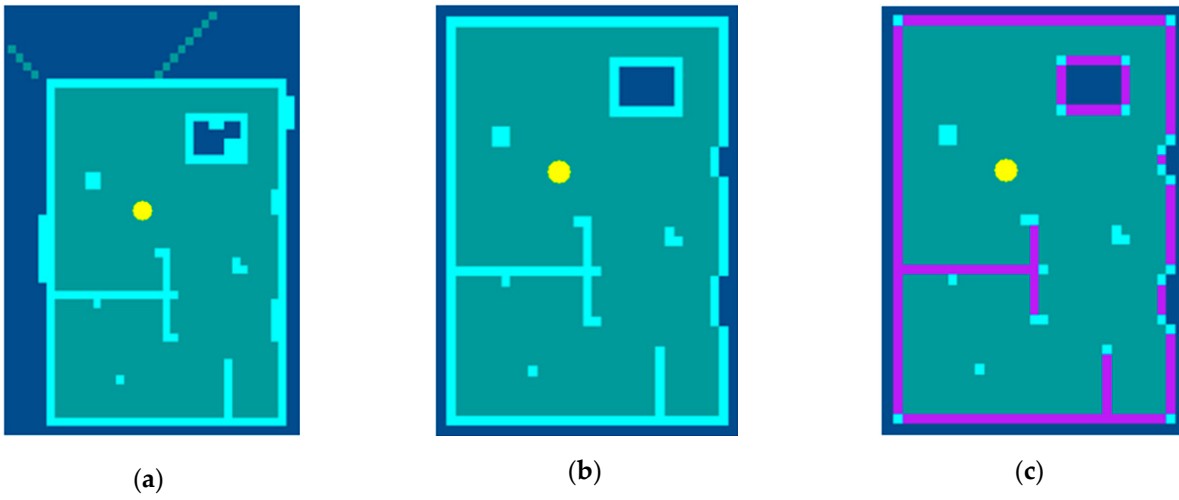

(**a**)          (**b**)          (**c**)

**Figure 5.** *Cont.*

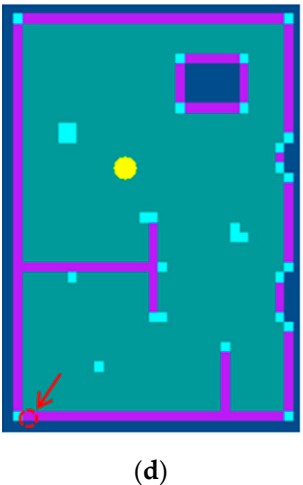 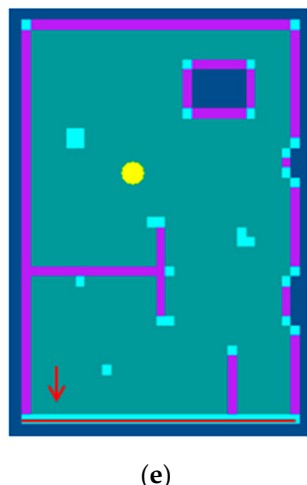 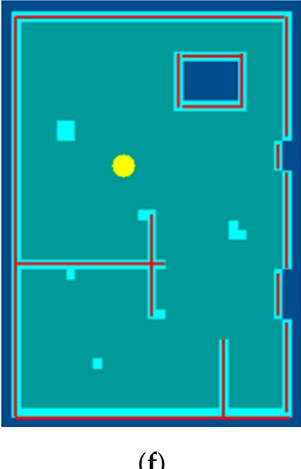

(**d**)　　　　　　　　　　　(**e**)　　　　　　　　　　　(**f**)

**Figure 5.** The process of RM-Line: (**a**) initial map; (**b**) edge map; (**c**) active map; (**d**) pitch an active point; (**e**) obtain a line segment; (**f**) line segments. The red lines are segments extracted from the map.

(5) As shown in Figure 5d, if $p(x,y)$ is an active point and marked as free, then go to (6); otherwise, go to (8).

(6) Grid cluster $\{p_0, p_1, \ldots, p_N\}$ is obtained using a method based on the ray model, and then fitted to a straight line using the method based on the least-squares, as shown in Figure 5e.

(7) Mark the active points $\{p_0, p_1, \ldots, p_N\}$ as locks.

(8) If $x < xsize$, then $x$++; otherwise, $x = 0$, $y$++.

(9) Repeat (4)–(8) until all edge points are processed. The straight-line extraction result is shown in Figure 5f.

## 3. Results

This paper implemented the proposed RM-Line on an indoor mobile robot, as shown in Figure 6. The robot collects environmental information and builds a grid map with the help of gyroscopes, odometers, laser scanners, collision sensors, etc. The radius of the robot is 17 cm, and the moving speed is about 25 cm/s. The measurement error of the gyroscope is $\pm 10^{\circ}$@60 min. The scanning frequency of the laser scanner is 2 Hz, the measurement range is 15–300 cm, and the measurement accuracy is 5 cm@200 cm. The robot has an embedded controller installed, with a 128 MByte DDR and 300 MHz running frequency.

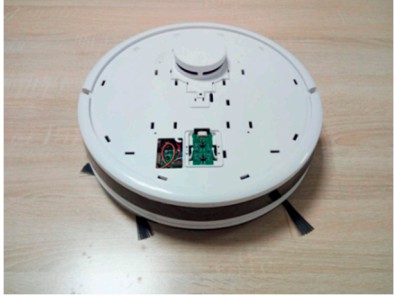

**Figure 6.** The mobile robot.

We tested RM-Line in several different environments to verify the effectiveness of the algorithm in different environments. Meanwhile, RM-Line was compared with Canny-Line. To the best of our knowledge, CannyLine is a state-of-the-art and well-known line extraction algorithm.

### 3.1. Evaluation Standard

We quantitatively evaluate the performance of the algorithm using three parameters: the number of lines, the average distance from grids to the lines, and the running time.

(1) Number of lines

One of the reasons why the number of lines is used as an evaluation parameter is because the robot needs to compress the map and reload it.

As shown in Figure 7a, the first time a robot works in an environment, a grid map is built. As shown in Figure 7b, after the first job, the robot extracts line segments from the grid map and stores them. As shown in Figure 7c, when the robot works in this environment again, the stored line segments are reloaded, and a grid map is generated accordingly.

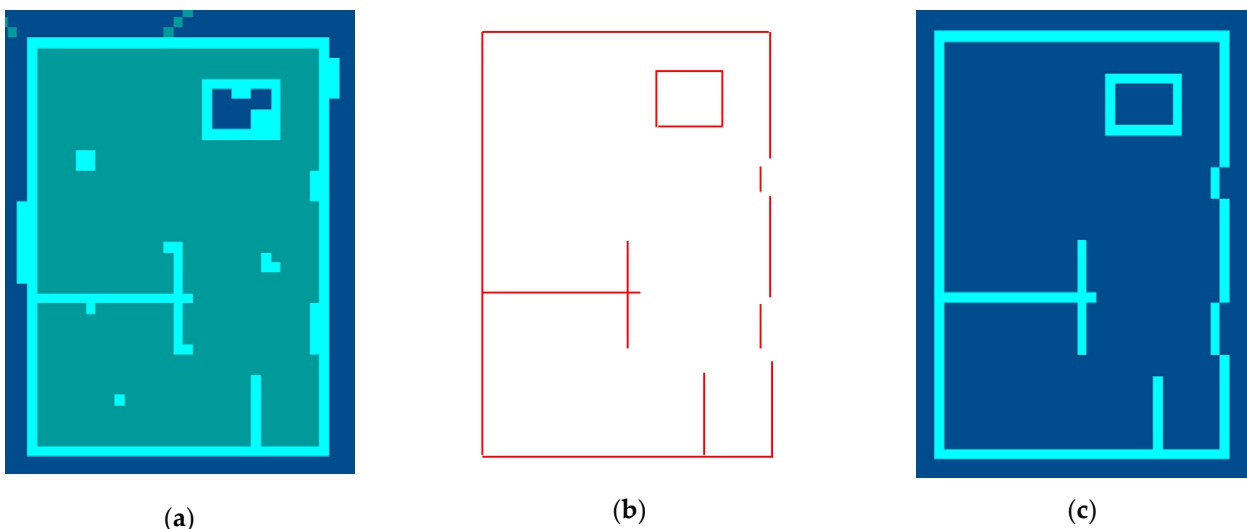

(**a**)  (**b**)  (**c**)

**Figure 7.** Compressed storage and reloading of the grid map: (**a**) origin map; (**b**) line segments; (**c**) map reloaded with lines.

It can be known from the storage and reloading process of the grid map that the number of line segments affects the storage space and efficiency.

(2) Average distance from grids to lines

From the storage and reloading process of the grid map, it can be known that the closer the extracted line segment is to the outline of the obstacle, the more accurate the reloaded map is. Therefore, the distance from grids to lines is used as an evaluation parameter.

The distance from grids to lines is shown in Figure 8, where L0–L5 are the straight lines obtained using the algorithm, the grass green grids are the obstacle grids in the edge map, and the black dots are their center. The black lines are the distance from the center of the grid to the nearest straight line. For example, the closest lint to grid Pi is L5, and the distance is di. The average point-to-line distance refers to the average of the distances from all grid points to the nearest line, as shown in Equation (5).

$$DisPL_{ave} = \frac{\sum_{n=0}^{N} d_n}{N},$$ (5)

where $N$ is the number of obstacle grids, and $d_n$ is the distance from grid $P_n$ to the nearest line.

The smaller the $DisPL_{ave}$, the closer the obtained straight line is to the real contour, i.e., the higher the accuracy.

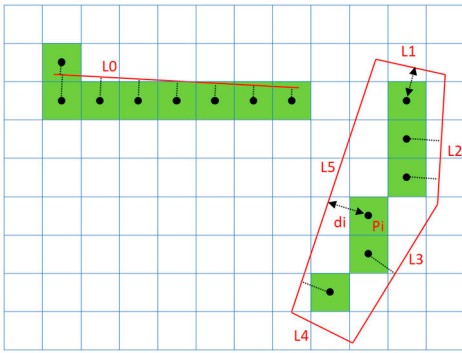

**Figure 8.** Distance from grids to lines.

## 3.2. Comparison Results in a Single-Room Environment

The first experiment was carried out in a separate room. Figure 9 shows the experimental results.

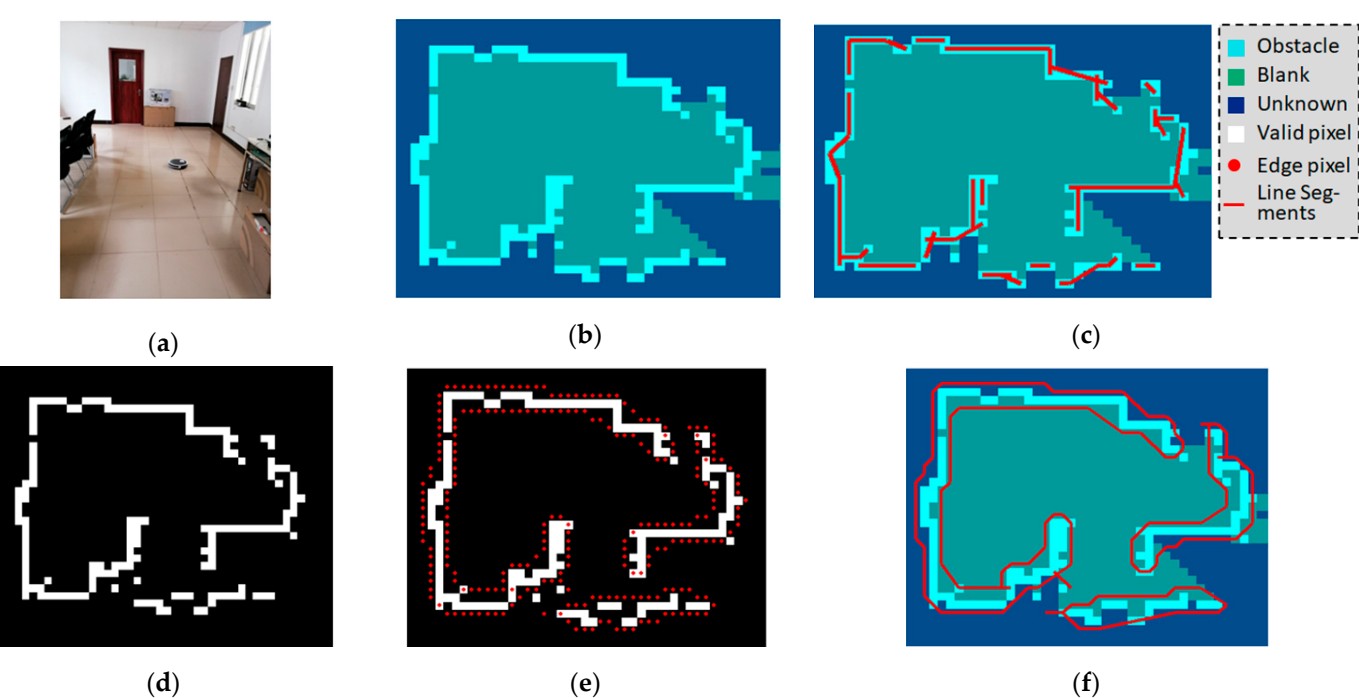

**Figure 9.** The experiment in a single room: (**a**) the environment; (**b**) grid map; (**c**) line segments obtained using RM-Line; (**d**) Canny-Line using grayscale image as input; (**e**) edge map obtained using Canny-Line; (**f**) line segments obtained using Canny-Line.

Figure 9a shows the experimental environment, which was a 5 m × 7 m office with some tables and chairs.

Figure 9b shows the grid map of the mobile robot. The size of each grid was 20 cm × 20 cm. The turquoise points are the BLANK grids, the cyan points are the OBSTACLE grids, and the blue points are the UNKNOWN grids.

Figure 9c shows the line segments obtained using the proposed RM-Line.

As shown in Figure 9d, before running CannyLine, the grid map was converted to a grayscale image, where the OBSTACLE grids had a gray value of 100 (white pixels in Figure 9d), and the BLANK grids and UNKNOWN grids had a gray value of 0 (black background in Figure 9d). Figure 9e shows the edge points obtained using CannyLine. Figure 9f shows the line segment obtained using CannyLine.

Table 1 shows the comparison of the two algorithms.

**Table 1.** Comparison of two algorithms in Experiment 1.

|  | Average Dis (cm) | Number of Lines | Time (ms) |
|---|---|---|---|
| CannyLine | 21.2 | 79 | 93 |
| RM-Line | 2.3 | 37 | 22 |

As shown in Table 1, RM-Line consumed less time than Canny, thanks to the use of ray models. Furthermore, RM-Line extracted 37 lines from the grid map, and CannyLine extracted 79 lines. When these lines were used for map compression, the proposed RM-Line saved 53% more storage space than CannyLine. More importantly, the straight lines obtained using CannyLine were distributed on both sides of the obstacle grid, which could not accurately describe the obstacle information in the map. Using them for map compression or map optimization would produce unexpectedly bad results. In contrast, the straight line obtained using RM-Line could accurately describe the obstacle information in the grid map, thereby meeting the needs of tasks such as map compression and map optimization of mobile robots.

*3.3. Comparison Results in Two-Room Environments*

Experiment 2 was performed in two rooms. Figure 10 shows the experimental results.

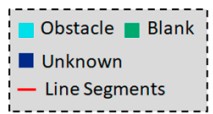

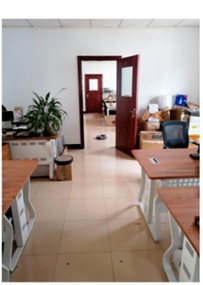

(**a**)

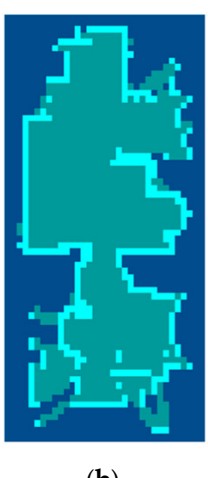

(**b**)

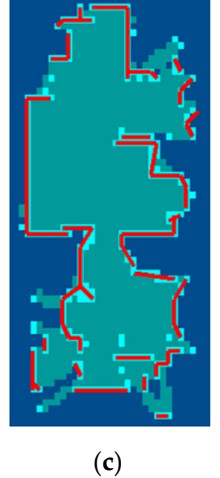

(**c**)

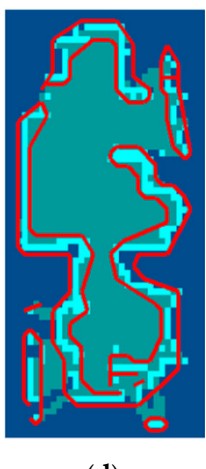

(**d**)

**Figure 10.** The experiment in two rooms: (**a**) the environment; (**b**) grid map; (**c**) line segments obtained using RM-Line; (**d**) line segments obtained using Canny-Line.

Figure 10a shows the experimental environment. It consisted of two rooms connected by a door. The environmental area was 5 m × 11 m. Compared with Experiment 1, the environment area of Experiment 2 was larger and had more obstacles.

Figure 10b shows the grid map of the robot, which had a larger extent than Figure 9b and contained more contours. The size of each grid was 20 cm × 20 cm.

Figure 10c shows the line segments obtained using RM-Line. It can be seen from the figure that, in a more complex grid map, RM-Line could also obtain an ideal straight line extraction effect.

Figure 10d shows the line segments obtained using CannyLine.

Table 2 shows the comparison of two algorithms.

**Table 2.** Comparison of two algorithms in experiment 2.

|  | Average Dis (cm) | Number of Lines | Time (ms) |
|---|---|---|---|
| CannyLine | 20.5 | 101 | 121 |
| RM-Line | 2.6 | 53 | 36 |

Table 2 shows that RM-Line saved 70% of the time compared to CannyLine, and the average point-to-line distance was reduced by 87%. RM-Line extracted 53 lines from the grid map, and CannyLine extracted 101 lines. When these lines were used for map compression, the proposed RM-Line saved 47.5% more storage space than CannyLine. Similar to the results in Section 3.2, the straight line obtained using CannyLine was quite different from the real obstacle grid, while the straight line obtained using the proposed RM-Line accurately described the obstacle information in the map.

### 3.4. Comparison Results in a Corridor Environment

Experiment 3 was performed in a corridor.

Figure 11a shows the experimental environment. It was a relatively regular rectangle, where the length was much larger than the width. There were doors at the end and sides of the corridor, but they all had a higher floor than the corridor; therefore, they were not represented in the robot's map. Compared with Experiment 2, the environment of Experiment 3 had fewer obstacles and the map structure was simpler. However, the long and narrow shape made most of the grids in the map contain straight lines, which brought different challenges for straight-line extraction.

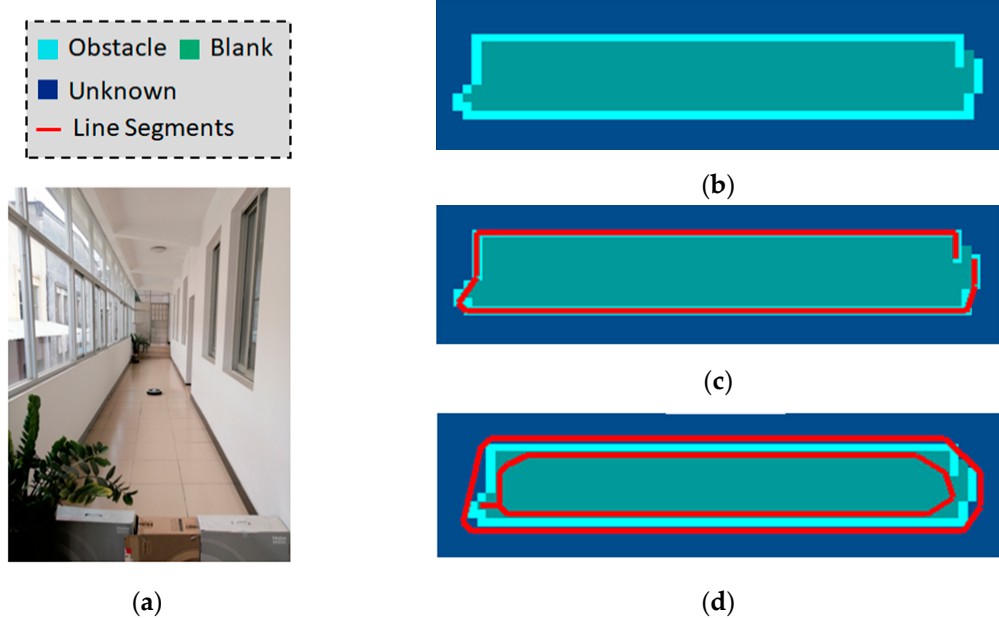

**Figure 11.** The experiment in a corridor: (**a**) the environment; (**b**) grid map; (**c**) line segments obtained using RM-Line; (**d**) line segments obtained using Canny-Line.

Figure 11b shows the grid map of the mobile robot. The size of each grid was 20 cm × 20 cm.

Figure 11c shows the line segments obtained using RM-Line, and Figure 11d shows the line segments obtained using CannyLine.

Table 3 shows the comparison of two algorithms.

**Table 3.** Comparison of two algorithms in experiment 3.

|  | Average Dis (cm) | Number of Lines | Time (ms) |
|---|---|---|---|
| CannyLine | 20.8 | 17 | 60 |
| RM-Line | 1.6 | 8 | 12 |

As shown in Figure 11, in a map containing long straight lines, RM-Line extracted accurate line segments, adequately reflecting the straight-line information in the map. CannyLine also obtained regular line segments, but they were distributed on both sides

of the real contours, which were not conducive for further data processing by the robot. RM-Line extracted eight straight lines from the grid map, while CannyLine extracted 17 straight lines. When these lines were used for map compression, the proposed RM-Line saved 52.9% more storage space than CannyLine.

## 4. Conclusions

This paper proposed a line segment extraction algorithm suitable for indoor mobile robot grid maps, called RM-Line. The algorithm uses the grid map's connected domain, i.e., the area formed by the connected blank grids, to find the obstacle grids located at the edge of the connected domain. On this basis, a scanning model is used to pick out active points, i.e., obstacle points that may contain complete line information. Then, a ray model was proposed, and the active points were used as the starting point of the ray model to obtain line segments in the grid map.

In the experimental phase, this paper conducted comparative tests on the proposed RM-Line and the state-of-the-art CannyLine in grid maps of three different environments. The experimental results show that, for the grid map of mobile robots, the proposed RM-Line could obtain more accurate line segments and consume less time.

This version of the algorithm only deals with line extraction on a 2D plane. In the next stage, we will try to extend the algorithm to 3D maps.

**Author Contributions:** Conceptualization, Y.Z.; methodology, H.L.; software, H.L. All authors read and agreed to the published version of the manuscript.

**Funding:** This research received no external funding.

**Institutional Review Board Statement:** Not applicable.

**Informed Consent Statement:** Not applicable.

**Conflicts of Interest:** The authors declare no conflict of interest.

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
