# Peer review of "RM-Line: A Ray-Model-Based Straight-Line Extraction Method for the Grid Map of Mobile Robot"

_applsci, doi:10.3390/app12199754_

Round 1
Reviewer 1 Report
See the reviewer's comments file.

Author Response
Thank you for your suggestion. We have made major revisions to the manuscript based on your suggestions.
Reviewer 2 Report
The author(s) did good work introducing a ray-model-based straight line extraction method for the grid map of mobile robot. However, the manuscript is insufficient to be accepted. revising the following points may enhanced the paper’s contribution.
· Introduction section is incomplete to address the problem and to outline the goals and the solution.
· The related works in very less than expected and it’s better to present the recent related works in a table form contains the performance metrics and their differences.
· Rewrite the formulas (equations 1-4) in an accurate way.
· Results not involve the performance metrics in terms of error(s) and accuracy?
· A table showing your system against the recent related ones is required.
· Conclusion section is not sufficient to concludes the manuscript and outlines the futuristic plan.
Based on mentioned points I regret to reject the paper,
Author Response

(The authors gave the same response as above.)

Reviewer 3 Report
Recommendations, notations, formal mistakes and questions:
Lines 10, 11 and 12: not correspondent with the template. Can be moved to the abstract.
Line 20: "... fast and accurate ...": This is not proved in the article.
Figure 1.: Pictures are in poor quality.
From Line 109 (many times, also in the text): Every variables and indexes must be written in italic font. Only numerical parts use normal font. (Recommend to use the equation editor.)
Line 155: Write a proper symbol for multiplication, the asterisk (*) has another meaning in math.
Equations 3 and 4: The variables are not explained. Please explain.
Figure 5: Is this figures from real data? How much distance does one pixel represent?
Line 183: Explain the term "connected domain", "active points",...
Line 198: Which sensors are really uses? What is the accuracy and repeatability of this sensors?
Line 264: Conclusion must be extended: Please describe the obtained solution in detail.
Figure 9: The real environment contains a door (alcove is visible on the picture (a) ), but your map is missing this information. Why?
Notation and questions: The reason (purpose) of the line extraction and practical use in mobile robotics is not sufficiently explained.
You wrote: ...when the data is missing more, some line segments will be lost.. : is this algorithm usable in practice?
How long does the data processing take? What computing power does your platform provide?
Author Response

(The authors gave the same response as above.)

Round 2
Reviewer 2 Report
The work looks fine however the paper structure, the results, performance are not perfectly presented.
The structure allows the paper to be read at several different levels. For example, many people skim Titles to find out what information is available on a subject. Others may read only titles and Abstracts. Our main goal here is to insure that at whatever level a person reads your paper, they will likely get the key results and conclusions. In addition to this, I was very keen to see the performance metric evaluation in terms of error and accuracy? In the Introduction section, the authors were required to add a paragraph detailing the paper's organization.
Author Response
- We have revised the abstract to introduce the algorithm process, evaluation criteria, and experimental results.
- We added a section to Chapter 1 to describe the structure of the article.
- We evaluate the algorithm with three parameters. They are introduced in the Abstract and Experimental sections.
Reviewer 3 Report
Thank you for accepting comments and editing the article, although it could still be improved.
Author Response
Thanks again.
I added a description of the evaluation method to the summary section